# Speech-XL: Towards Long-Form Speech Understanding in Large Speech Language Models

## Abstract

Despite the growing success of Large Speech Language Models (LSLMs) in processing short-term acoustic signals, their extension to long-form audio understanding is severely bottlenecked. This limitation stems from the limited context length and the exorbitant memory footprints required for long-form inference. In this work, we propose Speech-XL, a new model that capitalizes on the intrinsic key-value (KV) sparsification capacity of Large Language Models (LLMs) to achieve high-ratio speech input compression. Specifically, we introduce a novel special token, the Speech Summarization Token (SST), for each speech interval to encapsulate the intra-interval speech information into its associated KV pairs. The SST module is trained via instruction fine-tuning, employing a curriculum learning strategy where the SST learns to compress information in a progressive manner—advancing from low-ratio (simple) to high-ratio (challenging) compression. Despite utilizing significantly less training data than other baselines, our model achieves highly competitive performance on major benchmarks, including LongSpeech and AUDIOMARATHON. By addressing the long-standing bottlenecks in long-form audio modeling, our approach offers a novel perspective on the condensation of extensive acoustic sequences.

## 1. Introduction

Recent progress in Large Language Models (LLMs) (Touvron et al., 2023; Nie et al., 2025; Yang et al., 2025) has changed how we approach speech processing, leading to the development of Large Speech Language Models (LSLMs). By connecting audio encoders with the reasoning power of LLMs, models like Whisper (Radford et al., 2023), Qwen-Audio (Chu et al., 2023), and Audio-Flamingo (Kong et al., 2024) have performed exceptionally well in tasks like speech recognition and translation. However, while these models work great for short audio clips (usually under 30 seconds), they struggle significantly with long-form audio.

The main difficulty is the mismatch between speech sequence length and the computational profile of Transformer-based architectures (Vaswani et al., 2017). For the same semantic content, speech representations are typically several times longer than their textual counterparts, due to high frame rates and redundant acoustic structure. Self-attention incurs quadratic complexity in sequence length, and the associated key-value (KV) cache grows linearly, leading to prohibitive memory usage and inference latency as audio duration increases. In practice, most existing LSLMs are constrained to clips shorter than roughly 30 seconds, and naïvely extending them to long-form speech is computationally challenging.

A natural workaround is to adopt a cascaded pipeline, first transcribing speech into text and then applying a text LLM with a larger context window. However, such cascaded systems (Zhang et al., 2023a; Chu et al., 2024) introduce an information bottleneck: the intermediate textual representation discards paralinguistic and acoustic cues—such as emotion, prosody, and environmental context—that are often crucial for holistic understanding. End-to-end (E2E) LSLMs (Zhang et al., 2023a; Défossez et al., 2024) avoid this bottleneck by directly conditioning on continuous features or discrete audio tokens (e.g., SoundStream (Zeghidour et al., 2021)), but they still operate on dense, high-resolution speech sequences. As a result, current E2E models remain effectively short-context, and long-form speech understanding is underexplored.

In this work, we study how to scale E2E LSLMs to long-form speech without reverting to a purely textual intermediate representation. Our approach is motivated by a simple observation: speech representations are highly redundant and attention in LLMs is typically sparse. Neighboring frames often encode similar content, so they populate the KV cache with many near-duplicate entries, while attention maps reveal that only a small subset of tokens is queried fre-

[1]Anonymous Institution, Anonymous City, Anonymous Region, Anonymous Country. Correspondence to: Anonymous Author <anon.email@domain.com>.

Preliminary work. Under review by the International Conference on Machine Learning (ICML). Do not distribute.

quently. To exploit this redundancy, we introduce a *Speech Summarization Token* (SST), a specialized token that attends to a local temporal segment of the audio and produces a compact representation for that segment. After summarization, the model retains only the condensed KV entries associated with SSTs for subsequent layers and discards the KV entries of the original acoustic tokens, thereby shortening the effective sequence seen by the LSLM while preserving salient semantic and paralinguistic cues.

Simply training SSTs to operate at high compression ratios is non-trivial: aggressive compression can easily harm recognition accuracy and long-range reasoning. To make SST learning stable, we first build a strong speech-language backbone and then introduce a dedicated curriculum stage for SST. Concretely, *Speech-XL* is trained in a three-stage pipeline: we first align semantic representations via ASR-based pre-training, then perform instruction tuning following DIFFA-style synthetic data generation (Zhou et al., 2025), and finally apply an SST curriculum on long-form audio. In the curriculum stage, the model is initially trained under low compression ratios (e.g., $2\times$, $4\times$), where the information-retention problem is comparatively easy, and is then gradually exposed to higher compression ratios (e.g., $8\times$) and longer input sequences. This progressive schedule stabilizes the summarization behavior before the model is asked to compress ultra-long audio.

Our contributions are as follows:

- We formulate the problem of long-form end-to-end speech modeling for LSLMs and show that naïve scaling is fundamentally limited by sequence length and KV cache growth in Transformer architectures.

- We introduce Speech-XL, a framework that integrates Speech Summarization Tokens (SSTs) into LSLMs to perform learnable KV-space compression of speech representations, substantially reducing memory and computation for long-form audio while maintaining task performance.

- We propose a curriculum learning strategy that gradually increases the SST compression ratio, enabling stable training from short segments to multi-minute audio.

- We will release the inference code for Speech-XL to facilitate further research on long-context speech-language models.

## 2. Related Work

### 2.1. Large Speech Language Models

The rapid evolution of LLMs has concurrently catalyzed significant breakthroughs in the domain of LSLMs. By inherit-

ing the sophisticated reasoning and generative capabilities of LLMs, LSLMs (Wang et al.; Fang et al.; Zeng et al., 2024) have undergone an accelerated transformation, expanding their scope from simple transcription to complex multimodal understanding. Currently, the prevailing methodologies in the LSLM landscape can be categorized into two primary paradigms. The first paradigm, represented by models (Tang et al.; Ghosh et al.) such as Qwen-Audio (Chu et al., 2023) and Qwen2-Audio (Chu et al., 2024), typically employs an architectural pipeline consisting of a pre-trained speech encoder, a bridge adapter, and a LLM.

The second paradigm, exemplified by frameworks such as SpeechGPT (Zhang et al., 2023a) and AudioLM (Borsos et al., 2023), treats speech as a discrete linguistic sequence rather than a continuous signal. This approach typically leverages neural audio codecs—such as EnCodec (Défossez et al., 2022) or SoundStream (Zeghidour et al., 2021)—to discretize raw waveforms into a sequence of acoustic tokens. By mapping these tokens into the same vocabulary space as textual units, the LLM can perform unified generative modeling via an auto-regressive objective. Notwithstanding their impressive achievements, this body of work remains predominantly tethered to short-duration audio segments. To date, the vast landscape of long-form audio understanding remains largely under-explored.

### 2.2. Token Compression

Transformer-based models are inherently constrained by the memory and computational demands of processing extended contexts. In the fields of natural language processing and computer vision, researchers typically employ key-value cache eviction and token pruning as primary optimization strategies to alleviate these bottlenecks. From the perspective of KV cache optimization, Zhang et al. (2023b) proposed the Heavy Hitter Oracle (H2O), which maintains a dynamic balance between recent tokens and Heavy Hitter (H2) Tokens. The foundational insight of H2 is that a small fraction of tokens—termed Heavy Hitters—contribute the majority of the value when computing attention scores. Li et al. (2024) proposed SnapKV, which automatically compresses KV caches by selecting clustered important KV positions for each attention head. Xiao et al. introduced StreamingLLM, a framework that enables LLMs trained with finite-length attention windows to generalize to infinite sequence lengths without requiring any fine-tuning. The pivotal contribution of this work is the identification of "Attention Sinks"—the phenomenon where initial tokens capture a disproportionate amount of attention weight, making their retention essential for maintaining model stability during extended sequence processing.

In the field of token pruning, Token Merging (ToME) (Bolya et al., 2023) and DynTok (Zhang et al., 2025) represent

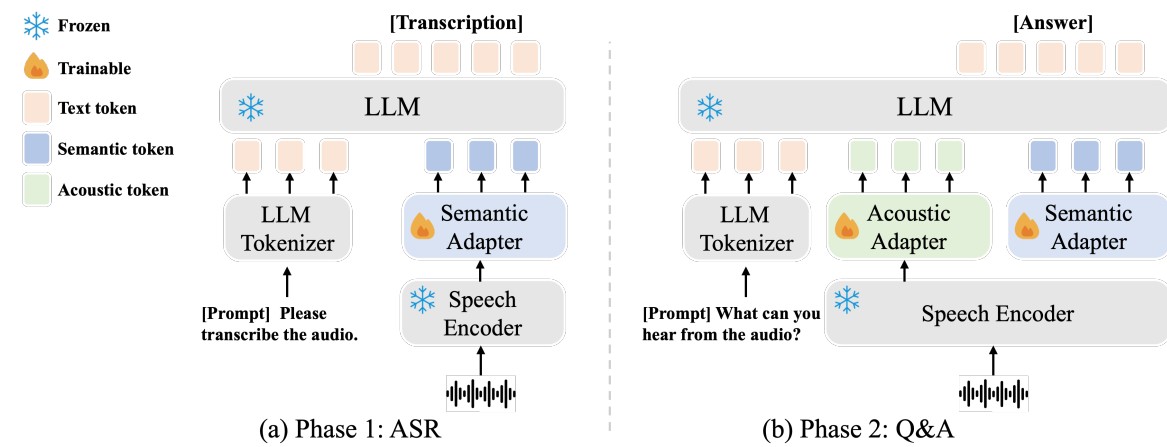

*Figure 1.* Training process of our Speech-XL framework. Phase 1: Semantic Alignment, which utilizes ASR tasks to bridge speech features and textual semantics. Phase 2: Modal Alignment, which employs speech captioning to refine the integration of speech context and linguistic descriptions.

two pivotal techniques for optimizing inference overhead. ToMe (Bolya et al., 2023) employs a bipartite matching algorithm to progressively consolidate similar tokens across layers, effectively reducing sequence length without the need for additional fine-tuning. In contrast, DynTok (Zhang et al., 2025) introduces a dynamic prediction mechanism that adaptively partitions tokens into groups and performs intra-group merging. This approach enables aggressive compression in regions of low information density while preserving essential representational fidelity in content-rich areas. While these methodologies have demonstrated efficacy in processing visual or textual tokens, research into speech token compression remains sparse, and the optimal strategy for transferring such techniques to the speech modality remains an open question.

## 3. Method

### 3.1. Overview

The Speech-XL framework is designed to empower LSLM to perceive and adapt to the complexities of long-form audio processing. The architecture facilitates this through a staged alignment process and a novel interval-based compression mechanism that leverages SST module.

### 3.2. Model Architecture

As illustrated in Fig.1 and Fig.2, the system consists of three primary components:

**Speech Encoder**: We employ Whisper-small as the core encoder, which can be configured as either a frozen or trainable module depending on the training phase. It extracts high-level acoustic features from raw waveforms to serve as the foundation for speech understanding.

**Dual-Adapter Bridge**: To bridge the gap between the speech encoder and the LLM's embedding space, we introduce a Dual-Adapter Bridge. This component projects encoder outputs into a latent space compatible with the LLM's dimensions through two specialized pathways:

Semantic Adapter: This adapter consists of a 2-layer convolutional network with a subsampling rate of 4, followed by a 2-layer linear projection.

Acoustic Adapter: This module utilizes 2-layer Q-former blocks equipped with 64 trainable query vectors. It specifically extracts acoustic-specific features from the intermediate states of the speech encoder, ensuring that nuances such as tone and environment are preserved.

**LLM with SST Compression**: We utilize Qwen2.5-7B-Instruct (Team et al., 2024) as our backbone Large Language Model, augmented with a specialized SST module. This module is designed to distill high-density, redundant speech tokens into compact KV states, effectively alleviating memory bottlenecks and enabling the model to process extended audio sequences.

### 3.3. SST Compression

We leverage the LLM's intrinsic representational power to generate compact speech embeddings. Given a sequence of speech tokens $\mathbf{X}$, we propose distilling the KV states of $\mathbf{X}$ into a condensed set of KV states for summarization tokens $\mathbf{C}$, where $|\mathbf{C}| \ll |\mathbf{X}|$. This strategy significantly reduces computational overhead and memory footprint, thereby enabling the model to accommodate substantially longer speech inputs.

**Compression mechanism**: When encoding a speech token $x_t$ within a long-form audio sequence $\mathbf{X}$, LLM must query

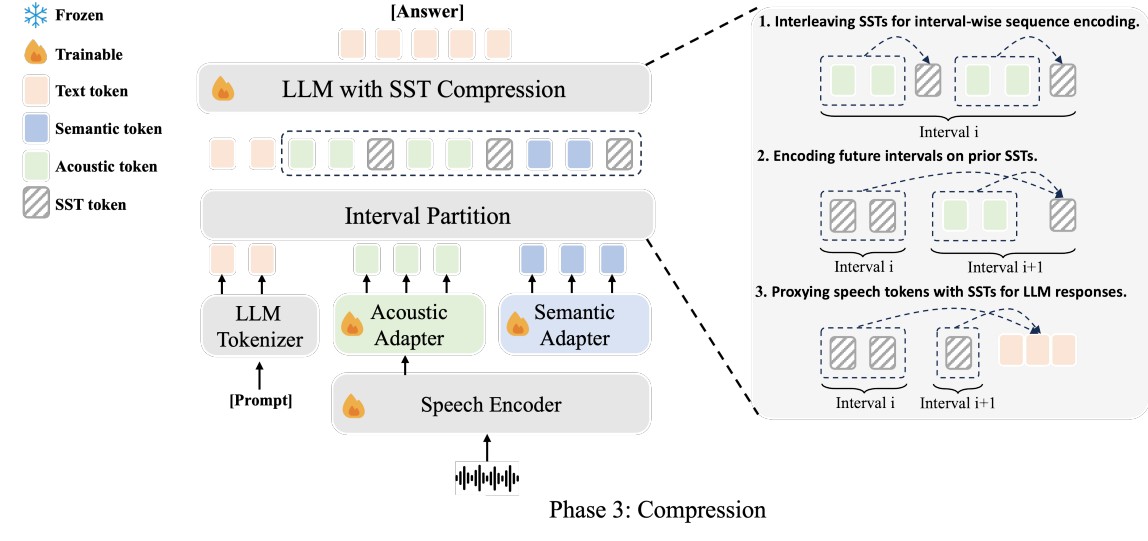

Phase 3: Compression

*Figure 2.* Training process of our Speech-XL framework. Phase 3: Long-form audio is encoded and partitioned into fixed intervals, where local information is distilled into SST KV states. This mechanism enables Speech-XL to perceive and comprehend exceptionally long speech streams.

the entire set of KV states from preceding tokens $\mathbf{X}_{<t}$. Consequently, this consumes significant GPU memory due to the storage of massive acoustic and semantic tokens, resulting in prohibitive computational costs due to the complexity of the self-attention mechanism. To avoid the immense cost of direct computation, we partition the audio feature sequence $\mathbf{X} = \{x_1, \ldots, x_n\}$ into shorter intervals $\{\mathbf{I}_1, \ldots, \mathbf{I}_N\}$ of sizes $\{w_1, \ldots, w_i\}$:

$$[x_1, \ldots, x_n] \xrightarrow{\text{Partition}} [\mathbf{I}_1, \ldots, \mathbf{I}_i], \qquad (1)$$

where $\sum w_i = n$ and $|\mathbf{I}_i| = w_i$. The length of each interval is 512.

For each interval, we introduce a new special token, the Speech Summarization Token (SST): $s$. This token prompts the LLM to compress the local acoustic and semantic information into the SST's KV states at every layer. We define a compression ratio $\alpha_i$ for each interval $\mathbf{I}_i$. Based on this ratio, we uniformly interleave $k_i$ SSTs into the interval (denoted as $\mathbf{S}_i = \{s_{i,1}, \ldots, s_{i,k_i}\}$), where $k_i = w_i/\alpha_i$. Specifically, one SST is appended for every $\alpha_i$ speech tokens:

$$\mathbf{I}_i \xrightarrow{\text{Interleave } \mathbf{S}_i} \mathbf{I}'_i, \qquad (2)$$

$$\mathbf{I}'_i = [x_{i,1}, \ldots, x_{i,\alpha_i}, s_{i,1}, \ldots, x_{i,\alpha_i+w_{i-1}}, s_{i,k_i}]. \qquad (3)$$

The LLM encodes these intervals sequentially. Once the encoding of interval $\mathbf{I}_i$ is complete, the SSTs' KV states $(\mathbf{S}_i)$ are preserved as the compressed representation of the audio information, while the high-density speech tokens' KV states $(\mathbf{I}_i)$ are off-loaded. When encoding the subsequent interval $\mathbf{I}_{i+1}$, the LLM directly conditions on the accumulated KV states from all preceding SSTs $(\mathbf{S}_{\leq i})$ as a proxy for the

original speech tokens $\mathbf{X}_{\leq i}$. This mechanism effectively bypasses the memory bottleneck, enabling Speech-XL to perceive and comprehend exceptionally long audio streams that would otherwise be computationally challenging.

**Curriculum Learning**: To further enhance the stability and representation efficiency of the model when processing long-form audio, we introduce a curriculum learning strategy for Speech-XL to dynamically adjust the compression ratio $\alpha$ during training. In the early stages of Phase 3 training, we initially set a lower compression ratio ($2\times, 4\times$) (i.e., retaining a larger number of SSTs), which provides the LLM with finer-grained acoustic and semantic anchors, thereby reducing the initial difficulty for the model to learn the information compression task. As training progresses, we expand the candidate pool for compression ratios (to include $8\times$), forcing the SSTs to extract key features across longer temporal spans and compress a broader range of redundant speech information into a minimal set of KV states. This easy-to-hard progression not only effectively prevents gradient collapse in the early stages of training but also ultimately empowers the model to perceive ultra-long audio streams at extremely high compression rates, ensuring the precise capture of core semantic information in long sequences while maintaining memory efficiency.

### 3.4. Training

**Stage 1**: ASR. The primary objective of the initial stage is to establish a robust mapping between acoustic signals and their corresponding linguistic meanings. We supervise this process using ASR task, where the model is prompted to transcribe audio segments. During this phase, optimization

is focused exclusively on the semantic adapter.

**Stage 2**: Q&A. Building upon the semantic foundation, the second stage aims to enrich the model's perception of paralinguistic and environmental nuances. We employ speech-based Question Answering to force the model to interpret complex audio contexts. In this stage, the acoustic adapter is optimized alongside the semantic adapter.

**Stage 3**: Compression. The final stage introduces the core SST compression mechanism to empower the framework with long-form audio understanding. The input sequence is partitioned into fixed-length intervals, where Speech Summarization Tokens (SSTs) are interleaved to distill local acoustic and semantic information into compact KV states. During this phase, the speech encoder, dual-adapter Bridge, and the LLM with SST Compression are jointly optimized end-to-end. This strategy forces the SSTs to act as efficient proxies for redundant speech tokens, drastically reducing the memory footprint.

Formally, given the compressed SST proxies from all intervals and the input instruction, the probability of generating the $(i+1)$-th token is formulated as:

$$P(t_{i+1} \mid \underbrace{p_1, \ldots, p_M}_{\text{prompt}}, \underbrace{\mathbf{S}_{1,1}, \ldots, \mathbf{S}_{N,k_N}}_{\text{compressed SST KVs}}, \underbrace{t_1, \ldots, t_i}_{\text{ground-truth}}; \Theta), \quad (4)$$

where $\mathbf{S}_{j,k}$ denotes the $k$-th SST from the $j$-th interval, and $\Theta$ represents the set of learnable parameters. We perform standard auto-regressive training by minimizing the negative log-likelihood (NLL) loss for the target sequence. The training loss $\mathcal{L}$ is defined as:

$$\mathcal{L}(\Theta) = -\sum_{i=1}^{L} \log P(t_i \mid \mathbf{S}_{1:N}, p_{1:M}, t_{<i}; \Theta). \quad (5)$$

## 4. Experimental Setup

### 4.1. Implementation

Speech-XL is built upon the Qwen2.5-7B-Instruct backbone and trained using a three-stage strategy. In stage 1, we align the speech embeddings from the frozen whisper-small encoder with the LLM's text embedding space by optimizing the semantic adapter. During stage 2, the parameters of both the semantic and acoustic adapters are jointly fine-tuned to capture nuanced speech features. In the final stage, we introduce the SST module for long-context understanding and perform full-parameter fine-tuning on the entire framework—including the speech encoder, dual-adapter, and the LLM. All experiments are conducted on a server equipped with $8 \times$ NVIDIA A800-80GB GPUs. The batch sizes for the three stages are set to 4, 2, and 1 per GPU, respectively. We apply learning rates of $5 \times 10^{-5}$ for stage 1, $1 \times 10^{-4}$ for stage 2, and $1 \times 10^{-5}$ for the final process.

### 4.2. Datasets

We employ a multi-stage data curation strategy to progressively build the model's linguistic foundation, acoustic sensitivity, and long-context reasoning capabilities. In stage 1, we utilize a subset of the Emilia dataset (He et al., 2024), comprising approximately 10,000 hours of English speech, to perform large-scale Automatic Speech Recognition (ASR) alignment. This ensures a robust mapping between the Whisper-small encoder's features and the LLM's linguistic embedding space. Building on this semantic foundation, stage 2 incorporates a diverse mixture of five specialized datasets totaling 127 hours—including VCTK-Corpus (Yamagishi et al., 2019), Accentdb (Ahamad et al., 2020), IEMOCAP (Busso et al., 2008), DailyTalk (Lee et al., 2023), and VoxCeleb1 (Nagrani et al., 2020) to refine the model's perception of nuanced paralinguistic cues. Finally, in stage 3, we leverage the training split of the LongSpeech (Yang et al., 2026) dataset to cultivate the model's capability for long-form audio context compression, empowering the SST mechanism to capture long-range dependencies across extended temporal spans.

### 4.3. Benchmarks

**LongSpeech** (Yang et al., 2026) serves as a rigorous evaluation framework specifically targeting long-context speech understanding, with audio durations extending up to 10 minutes. The benchmark encompasses challenging tasks such as long-form speech summarization, key information retrieval, and long-range logical reasoning, which test a model's ability to capture dependencies over massive context windows.

**AudioMarathon** (He et al., 2025) is a comprehensive benchmark designed to evaluate the multi-task understanding and reasoning capabilities of models across audio segments ranging from 1 to 5 minutes. It categorizes 10 distinct sub-tasks into three primary dimensions: Speech Content Extraction (including ASR, SCR, and SER), Audio Classification (SED, MC, and ASC), and Speaker Information Modeling (SD, ER, SAR, and SGR). Unlike traditional datasets that focus on isolated phonetic or semantic tasks, AudioMarathon requires models to maintain fine-grained acoustic perception while performing high-level semantic reasoning over mid-length temporal spans, thereby providing a holistic measure of a model's robustness in complex, real-world multi-modal scenarios.

## 5. Experiments

### 5.1. In-domain Evaluation

The experimental outcomes are summarized in Table 1 and Table 2. In the LongSpeech evaluation, Speech-XL demonstrates a significant advantage in most tasks. It is necessary to clarify that Speech-XL undergoes fine-tuning on this spe-

*Table 1.* Performance Comparison on the LongSpeech Dataset including Content Separation, Emo. (Emotion Analysis), Speaker Count, Summary, Lang. Det. (Language Detection), and Temp. Loc. (Temporal Issue Localization). Metric abbreviations are as follows: N.A (Numeric Accuracy), Pr (Post-Parsing Precision), St.A (Strict Accuracy), R.A (Relaxed Accuracy), R1 (ROUGE-1 F1), R2 (ROUGE-2 F1), RL (ROUGE-L F1), D.A (Detection Accuracy), and D.E (Detection Errors). Best results are in **bold**.

| Model | Content Separation | | Emo. | | Speaker Count | | Summary | | | Temp. Loc. | |
|---|---|---|---|---|---|---|---|---|---|---|---|
| | N.A | Pr | St.A | R.A | N.A | Pr | R1 | R2 | RL | St.A | R.A |
| AudioFlamingo3(Goel et al., 2025) | 3.33 | 5.03 | 18.53 | 34.13 | 21.62 | 30.61 | 20.25 | 4.92 | 12.97 | 6.10 | 12.52 |
| Voxtral(Liu et al., 2025) | 25.73 | 33.85 | 38.80 | 51.62 | 28.50 | 28.57 | 41.81 | 14.61 | 25.10 | **23.69** | **48.81** |
| DashengLM(Dinkel et al., 2025) | 23.75 | 23.98 | 11.08 | 29.53 | 35.31 | 35.31 | 15.22 | 1.24 | 10.38 | 0.48 | 6.10 |
| Speech-XL | **72.84** | **72.84** | **51.94** | **66.98** | **84.72** | **84.72** | **49.72** | **15.07** | **26.76** | 16.09 | 32.96 |
| Upper-bound | 77.28 | - | 57.96 | 72.05 | 87.89 | - | 52.72 | 17.55 | 29.33 | 16.73 | 43.82 |

*Table 2.* Speech Recognition Performance on the LongSpeech dataset. Best results are in **bold**.

| Model | WER |
|---|---|
| Whisper(Radford et al., 2023) | 17.0 |
| Kimi-audio(Ding et al., 2025) | 51.2 |
| AudioFlamingo3(Goel et al., 2025) | 34.5 |
| Voxtral(Liu et al., 2025) | 27.5 |
| DashengLM(Dinkel et al., 2025) | 35.5 |
| Qwen2-Audio(Chu et al., 2024) | 29.9 |
| Speech-XL | **11.4** |
| Upper-bound | 11.0 |

cific dataset, while the compared models operate primarily in a zero-shot setting.

To preliminarily assess information loss in the SST mechanism, we compare the compressed Speech-XL with an uncompressed upper-bound model. Experiments reveal minimal performance gap between them. Combined with the low Word Error Rate (WER) in Table 2, this proves that the mechanism preserves rich acoustic and semantic information while significantly compressing the sequence. Unlike compared models that suffer from attention dispersion and context loss in long sequences, Speech-XL transforms redundant features into high-density representations via the SST mechanism. This not only shortens the computational sequence but also ensures stable understanding of long-form speech, fully validating the superiority and effectiveness of the SST mechanism.

In summary, the results on the LongSpeech benchmark validate the representation capacity of the SST mechanism. To further assess the generalizability and robustness of this mechanism across unseen domains and diverse tasks—and to verify whether the model has truly mastered the generalized paradigm of recursive compression—we provide a in-depth analysis in the out-of-domain evaluation section.

## 5.2. Out-of-domain Evaluation

To assess the long-form audio perception and reasoning abilities of our compression-based model, we evaluate Speech-XL on the AudioMarathon benchmark, which assesses multifaceted speech understanding across content extraction, classification, and speaker modeling dimensions.

Firstly, Table 3 indicates that while Speech-XL's total score (48.9) lags behind SOTA models like Qwen2.5-Omni (70.5), our Upper-bound (51.6) also fails to surpass these models. This phenomenon indicates that the gap between Speech-XL and top open-source/commercial models fundamentally stems from the order-of-magnitude difference in foundational model pre-training data volume (10k hours vs. hundreds of thousands/millions of hours), rather than losses incurred by compression mechanisms. In other words, the SST architecture has reached the current capability ceiling of foundational models at an extremely low computational cost, validating its effectiveness. It holds significant scalability potential as foundational models continue to advance.

Secondly, since our training data primarily focuses on human speech and does not encompass non-speech audio content, we concentrate our analysis on two core tasks: Speech Content Extraction (SCE) and Speaker Information Modeling (SIM). The experimental results are shown in Figure 3. Regarding SCE, Speech-XL approaches the performance bottleneck, with its score of 67.6 (ranking 3rd) differing from the Upper-bound by merely 1.2%. This performance not only validates the precision of its semantic extraction but, more importantly, demonstrates the SST mechanism's ability to ensure critical information capture remains unaffected by noise through effective redundancy compression.

Additionally, Speech-XL demonstrates promising performance on the SIM dimension. Despite training on only 127 hours of audio data, its scores remain at the mid-range level, with a gap of just 1.4% from the upper bound. This indicates that Speech-XL's feature space rapidly converges to key representations of speaker identity, validating its sig-

*Table 3.* Performance comparison of models on AudioMarathon dataset. Best scores are in **bold**.

| Models | Speech Content Extraction | | | Speaker Information Modeling | | | | Avg | Audio Classification | | | Overall |
|---|---|---|---|---|---|---|---|---|---|---|---|---|
| | SER | SCR | ASR | SD | ER | SAR | SGR | | SED | MC | ASC | |
| Qwen2.5-Omni-7B (Xu et al., 2025) | 26.3 | **85.1** | **98.1** | **72.3** | 53.4 | 21.4 | 98.0 | **65.6** | 78.4 | 100.0 | **72.2** | **70.5** |
| Qwen2.5-Omni-3B (Xu et al., 2025) | 25.2 | 82.3 | 94.7 | 67.3 | 39.6 | 29.1 | 97.2 | 62.9 | 70.2 | 97.4 | 69.3 | 67.2 |
| Audio-Flamingo-3(Goel et al., 2025) | 21.7 | 78.9 | 94.3 | 33.7 | **54.3** | **40.7** | 96.2 | 60.6 | 59.5 | 97.0 | 54.1 | 63.0 |
| Gemini-2.5-Flash (Comanici et al., 2025) | 28.1 | 83.6 | 96.8 | 33.1 | 31.9 | 34.3 | **99.3** | 59.6 | 69.2 | 79.3 | 40.8 | 59.6 |
| Voxtral-Mini-3B-2507(Liu et al., 2025) | 24.3 | 71.1 | 96.8 | 68.0 | 29.7 | 30.7 | 71.0 | 57.0 | 71.0 | 83.8 | 27.2 | 57.4 |
| **Upper-bound** | - | - | - | - | - | - | - | 56.1 | - | - | - | 51.6 |
| Gemini-2.5-Flash-Lite (Comanici et al., 2025) | 30.3 | 64.0 | 96.5 | 33.9 | 14.6 | 19.6 | 77.9 | 50.1 | 68.0 | 64.8 | 36.8 | 50.6 |
| Gemma-3n-E4B-it (Team, 2025) | 19.0 | 56.9 | 93.2 | 35.9 | 18.9 | 21.8 | 93.0 | 49.4 | 50.2 | 71.9 | 31.7 | 49.3 |
| **Speech-XL** | 31.1 | 75.6 | 96.1 | 33.3 | 30.3 | 26.6 | 77.8 | 54.7 | 53.4 | 40.4 | 24.3 | 48.9 |
| GPT-4o-Audio (2024-12-17) (Hurst et al., 2024) | 25.7 | 60.2 | 94.7 | 30.8 | 21.8 | 19.9 | 73.1 | 48.3 | 51.2 | 67.6 | 41.9 | 48.7 |
| Phi-4-Multimodal (Abdin et al., 2024) | 18.4 | 69.3 | 92.7 | 26.4 | 27.3 | 26.6 | 91.1 | 51.5 | 55.1 | 46.7 | 23.4 | 47.7 |
| GPT-4o-Audio (2024-10-01) (Hurst et al., 2024) | 25.8 | 61.4 | 94.4 | 32.5 | 22.5 | 17.2 | 69.2 | 48.0 | 50.7 | 59.5 | 40.8 | 47.4 |
| Gemma-3n-E2B-it (Team, 2025) | 22.5 | 51.6 | 91.3 | 35.1 | 15.2 | 12.2 | 91.6 | 46.8 | 50.2 | 56.8 | 28.2 | 45.5 |
| Aero-1-Audio (Li et al., 2025a) | 17.9 | 56.6 | 43.7 | 33.7 | 32.0 | 17.8 | 47.5 | 36.1 | 55.0 | 83.9 | 39.9 | 42.8 |
| Baichuan-Omni-1.5 (Li et al., 2025b) | 12.4 | 11.2 | 86.5 | 49.2 | 18.9 | 10.2 | 81.5 | 38.4 | 45.7 | 52.0 | 25.8 | 39.3 |
| Audio-Flamingo-2 (Ghosh et al., 2025) | 26.8 | 39.8 | 1.0 | 45.9 | 13.1 | 20.3 | 85.1 | 31.8 | 27.1 | 66.8 | 29.7 | 35.6 |

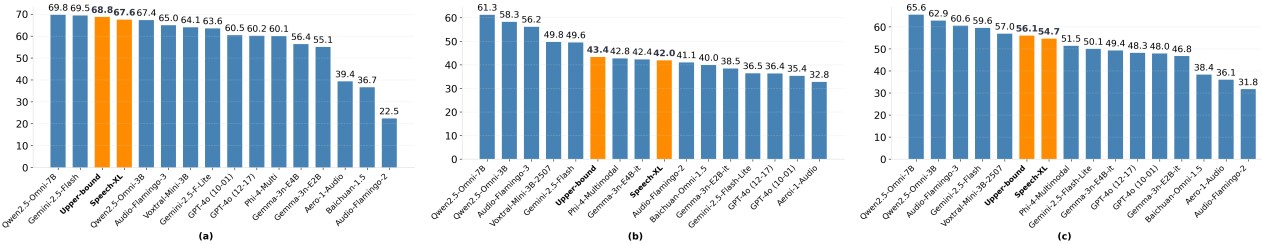

*Figure 3.* (a) shows the ranking for the Speech Content Extraction task. (b) shows the ranking for Speaker Information Modeling. (c) presents the overall ranking combining both tasks.

nificant potential for achieving high-performance speaker modeling in low-resource environments.

Finally, Speech-XL rank eighth overall. However, when audio categories not covered in the training data are excluded, Speech-XL's ranking jumped to sixth place. This performance improvement indicates that further enhancements to Speech-XL's capabilities are primarily constrained by the coverage of the training data. Future research will focus on enhancing the model's perception of fine-grained acoustic and prosodic cues, while incorporating more diverse audio corpora. This aims to ensure the SST module can fully capture subtle paralinguistic features beyond semantic information while maintaining high compression ratios.

### 5.3. Efficiency and Computational Overhead Analysis

Figure 4 intuitively demonstrates Speech-XL's advantages. As audio duration increases from one minute to ten minutes, Qwen2.5-Omni-7B exhibits sharp step-like growth in GPU memory consumption and computational overhead (TFLOPs), whereas Speech-XL's growth curve remains relatively smooth and gradual. During 10-minute long-form audio inference, Speech-XL's TFLOPs are only approximately 60% of its competitor's, and its memory consumption is also significantly lower. This proves that the SST

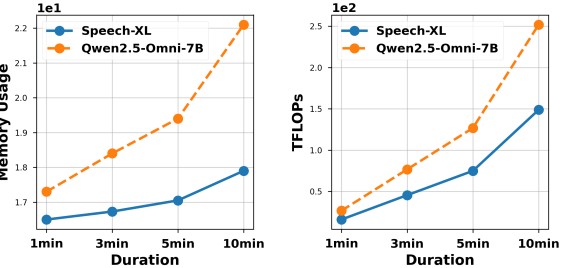

*Figure 4.* Comparison of the memory usage and the forward FLOPs between ours and Qwen2.5-Omni-7B.

mechanism, by compressing long audio into refined token sequences, not only addresses the attention dilution problem that models face with long sequences, but also achieves notable improvements in inference efficiency.

### 5.4. Ablation Study

To validate the model's effectiveness, we conduct a comprehensive ablation analysis of the compression mechanism and training strategy.

**Compression mechanism.** We conduct a systematic comparison between Speech-XL and mainstream compression

*Table 4.* Comparison of different compression techniques. SCE stands for Speech Content Extraction, SIM stands for Speaker Information Modeling, and AC stands for Audio Classification.

| Model | SCE | SIM | AC | Overall |
|---|---|---|---|---|
| Avg Pooling | 62.8 | 39.4 | 34.3 | 45.3 |
| Max Pooling | 53.6 | 40.6 | 34.0 | 42.5 |
| Similarity | 62.3 | 40.8 | 35.2 | 45.9 |
| Speech-XL | **67.4** | **41.3** | **38.4** | **48.3** |
| Upper-bound | 68.8 | 43.4 | 43.1 | 51.6 |

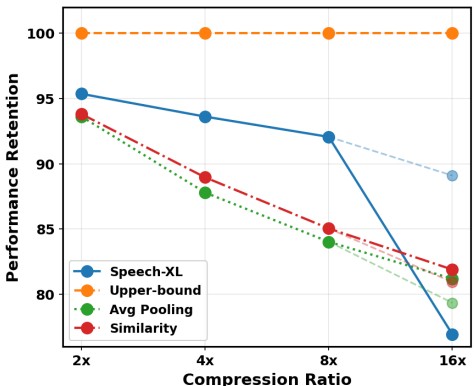

*Figure 5.* Performance of various compression methods across different compression ratios. Dashed line represents performance excluding ASR, while the solid line represents the overall performance. For the 16× ratio, Speech-XL is evaluated through direct inference without fine-tuning.

methods, including average pooling, max pooling, and similarity-based pruning (FastLongSpeech (Guo et al., 2025) / ToMe (Bolya et al., 2023) style) schemes. To ensure fair evaluation, all comparison methods are conducted under identical model architectures and training data, with a unified 4× compression ratio. The experiments are carried out on the AudioMarathon dataset, and the results are presented in Table 4. The experimental results demonstrate that Speech-XL exhibits absolute advantages on the core metric SCE, outperforming average pooling by 4.6% and similarity-based pruning by 5.1% respectively. Notably, at a 4× compression ratio, Speech-XL exhibits the smallest gap between its performance and upper bound, where the SCE metric differs by only 1.4%, and the overall score difference is merely 3.3%. This reflects that the SST mechanism, by enhancing the information density per token, not only retains the model's ability to capture fine-grained details such as word error rates, but also demonstrates ideal contextual integration capability on coarse-grained tasks such as long-form audio summarization.

Furthermore, Figure 5 explores model performance across various compression ratios. Since direct extrapolation to 16× causes a significant performance drop in ASR tasks,

*Table 5.* Evaluation of curriculum learning.

| Model | SCE | SIM | AC | Overall |
|---|---|---|---|---|
| w/o random ratio | 63.8 | 40.9 | 37.7 | 47.5 |
| w/o curriculum learn | 64.7 | 41.0 | **39.6** | 48.4 |
| Speech-XL | **67.6** | **42.0** | 39.4 | **48.9** |

we use two lines to visualize the results: dashed lines represent performance excluding ASR, while solid lines show overall performance. In experiments, Speech-XL maintains a marginal gap with the upper-bound, significantly outperforming the baselines. Specifically, the dashed lines demonstrate that the model effectively retains its performance even at an unseen compression ratio (16×), underscoring the generalization and robustness of the proposed method.

**Curriculum learning.** To explore the impact of curriculum learning strategy on model performance, we design two sets of ablation experiments: 1) w/o random ratio: we remove random sampling and consistently use a fixed 8× compression ratio throughout training; 2) w/o curriculum learn: we remove the process of progressively increasing compression intensity. The experimental results presented in Table 5 reveal the necessity of progressive learning from simple to complex. This strategy not only improves fine-grained semantic extraction capability (such as SCE), but also achieves superior performance in coarse-grained speaker modeling (SIM). This progressive training paradigm ensures that the SST mechanism can establish robust speech-semantic alignment, preventing the model from encountering task complexity that exceeds its current learning capacity when exposed to extreme compression ratios prematurely.

## 6. Conclusion

In this paper, we propose Speech-XL, an innovative framework specifically designed for long-form speech understanding, capable of efficient modeling based on dense compressed representations generated by speech summarization tokens. Specifically, to dramatically reduce sequence length while precisely preserving speech features, we introduce the SST compression mechanism, which significantly enhances the information density per token. Furthermore, to improve learning efficiency, we introduce a curriculum learning paradigm that guides the model to progressively master semantic alignment under extreme compression through smooth transitions from low to high compression ratios. Experimental results demonstrate that Speech-XL proves its effectiveness on both in-domain (LongSpeech) and out-of-domain (AudioMarathon) benchmarks, while delivering competitive compression quality and computational efficiency. Most importantly, by overcoming the long-standing bottleneck in modeling long-form speech, Speech-XL offers a fresh perspective on compressing large-scale sequences.

## Impact Statement

This paper presents work whose goal is to advance the field of long-form speech understanding. There are many potential societal consequences of our work, none which we feel must be specifically highlighted here.

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
