# OpenReview forum: "Speech-XL: Towards Long-Form Speech Understanding in Large Speech Language Models"
_ICML.cc/2026/Conference — Submitted to ICML 2026_

### Official Review · Reviewer_Fgkf · 2026-03-06

**Soundness:** 1
**Presentation:** 3
**Significance:** 2
**Originality:** 2
**Overall Recommendation:** 2
**Confidence:** 4

**Summary:**

This paper introduces Speech-XL, a model tailored for long-form speech understanding. To overcome the quadratic complexity of attention and the linear memory growth of the KV cache for extended audio, the authors propose a Speech Summarization Token (SST) mechanism. The input audio sequence is partitioned into fixed-length intervals, and SSTs are interleaved to compress local acoustic and semantic information into a reduced set of KV states. The model is trained via a three-stage pipeline, utilizing a curriculum learning strategy that progressively increases the compression ratio.

**Compliance With Llm Reviewing Policy:**

Affirmed.

**Final Justification:**

N/A

**Key Questions For Authors:**

1. Can you provide a fair comparison on LongSpeech where the baseline models are also fine-tuned on the dataset? Alternatively, how does Speech-XL perform in a strictly zero-shot setting on LongSpeech to isolate the true benefits of the SST mechanism?
2. Could you provide detailed diagnostic experiments that plot performance as a function of continuous audio length, and evaluate success rates across varying cross-interval dependency spans?
3. What specific information is retained by the SSTs? A mechanistic analysis showing whether SSTs encode lexical, prosodic, or speaker-identity information at different layers and compression ratios would greatly strengthen the paper.
4. Why was the interval length fixed at 512, and how sensitive is the model to this hyperparameter? Furthermore, how does the SST insertion compare to other long-context methods like Perceiver-style cross-attention bottlenecks or learned pooling? What are the distinct contributions of the semantic and acoustic adapters?

**Limitations:**

yes

**Strengths And Weaknesses:**

Strengths:
1. The proposed SST mechanism successfully reduces memory usage and TFLOPs during inference for long-form audio (as shown in Figure 4), providing practical resource savings.
2. The progressive curriculum learning strategy, transitioning from low-ratio to high-ratio compression, is an intuitive and empirically effective method to stabilize the learning of compressed representations.

Weaknesses:
1. The core architectural design is largely an application of existing LLM sequence compression and proxy/summary token techniques to the speech modality. It lacks substantial architectural innovation tailored specifically to the unique properties of continuous speech.
2. The central claim is that SST enables efficient long-form speech understanding without losing key information. However, the evaluation mainly showcases end-task benchmark scores rather than detailed diagnostic experiments. The paper lacks analysis on performance varying continuously with audio length, success rates across different dependency spans, or cross-interval information fidelity. The current results demonstrate good benchmark performance but fall short of definitively proving the mechanistic solution to long-context reasoning.
3. There is a critical fairness issue in the LongSpeech evaluation. Section 5.1 explicitly states that Speech-XL was fine-tuned on the LongSpeech dataset, while the baseline models were evaluated mostly in a zero-shot setting. This makes it impossible to disentangle how much of the performance advantage comes from the SST mechanism versus benchmark-specific adaptation, heavily undermining the claims in Table 1 and Table 2. Reviewers will immediately suspect a setup advantage rather than a methodical advantage.
4. The AudioMarathon results do not strongly prove the generalization benefits of the compression mechanism. The performance gap between Speech-XL and stronger models is attributed to the base model's pre-training data volume. However, without a controlled experiment isolating base model capacity from the compression mechanism, it remains unproven whether SST would scale effectively to stronger base models or if the advantage holds across different architectures. The upper-bound experiment merely shows that compression works well within their own constrained system.
5. The training data focuses exclusively on human speech, whereas the AudioMarathon benchmark includes non-speech audio classification (SED, MC, ASC). This mismatch confounds the interpretation of the overall score, leaving it ambiguous whether low performance on certain tasks is due to poor generalization of the compression model or an unfair distribution shift.
6. There is virtually no analysis on what information the SSTs actually retain. It is unclear whether the compression favors lexical, speaker, prosodic, or temporal boundary information. Furthermore, there is no discussion on how different layers behave, or what specific capacities drop first at higher compression ratios.
7. The ablations miss the most critical design variables. The paper needs to address: Why use SST insertion instead of Perceiver-style latent bottlenecks, learned pooling, or memory tokens? Why is the interval length fixed at 512, and is the model sensitive to this hyperparameter? What are the individual contributions of the semantic vs. acoustic adapters to long-context tasks? What happens if KV retention/dropping varies by layer?

---

### Official Review · Reviewer_TBRX · 2026-03-09

**Soundness:** 3
**Presentation:** 3
**Significance:** 2
**Originality:** 2
**Overall Recommendation:** 4
**Confidence:** 5

**Summary:**

This paper addresses long-form speech understanding in Large Speech Language Models (SLLMs). The authors propose Speech-XL, a framework designed to reduce speech sequence length and decoding costs through a token compression mechanism. The core approach involves using "Speech Summarization Tokens" (SST) to compress the Key-Value (KV) cache of adjacent intervals, allowing the model to handle longer audio inputs more efficiently. The authors demonstrate that the method achieves significant compression with acceptable performance degradation in accuracy.

**Compliance With Llm Reviewing Policy:**

Affirmed.

**Final Justification:**

N/A

**Key Questions For Authors:**

- Relation to Activation Beacon: How does the SST mechanism fundamentally differ from the Activation Beacon [1] approach? Could you provide a quantitative or qualitative comparison demonstrating why a speech-specific summarization token is necessary?

- Clarification of "Tokens": Can the authors clarify why the term "token" is used for continuous projector outputs? Is there any discretization occurring that was not mentioned? Also, what evidence supports the claim that the convolutional adapter is "semantic" while the Q-former is "acoustic"?

- Comparison with Downsampling: How does Speech-XL perform against a baseline that uses simple downsampling (e.g., 2x or 4x) at the encoder or projector level to reach the same compression ratio?

- Training Speed: Does the custom attention mask required for SST prevent the use of FlashAttention? Please provide a comparison of the actual training speed (wall-clock time) against standard SLLM architectures (using FlashAttention).

**Limitations:**

yes

**Strengths And Weaknesses:**

Strengths
- Timely Topic: Developing SLLMs capable of long-form speech understanding is a promising and highly relevant direction in the current research landscape.
- Efficiency: The proposed method successfully reduces the sequence length, which is a critical bottleneck for deploying SLLMs in real-world long-audio scenarios.
- Experimental Results: The results indicate that the trade-off between compression ratio and model accuracy is generally acceptable for the tasks evaluated.

Weaknesses
- Lack of Novelty and Comparison with Prior work: The proposed SST method appears highly similar to existing context compression techniques in the general ML field, specifically Activation Beacon [1]. Both methods use special tokens interleaved in the sequence to represent local KV cache intervals. Simply applying a known technique to the speech domain is insufficient for a claim of good originality; the authors must explicitly compare their work with such baselines and highlight the specific differences or speech-centric adaptations. Note [1] is only an example. The related work section should also be expanded to include this series of works.
- Confusing Technical Terminology (Soundness): The use of the terms "acoustic token" and "semantic token" (Line 189) is highly confusing. In speech literature, "acoustic tokens" usually refer to quantized units from codecs (e.g., EnCodec), and "semantic tokens" refer to clustered SSL units (e.g., HuBERT). However, in this paper, these terms are used to describe outputs from a convolutional-based projector and a Q-former-based projector, respectively. These are continuous representations, not discrete "tokens." Furthermore, the authors do not provide evidence that these two specific projectors successfully disentangle acoustic and semantic information as their naming suggests.
- Significance vs. Simple Baselines: Speech representations are inherently redundant. The authors have not compared their complex compression strategy against simpler, effective alternatives such as progressive downsampling in the encoder (e.g., Efficient Conformer) or simple striding in the projector. If a simple downsampling baseline can achieve similar performance, the utility of the proposed method is diminished.
- Hardware Compatibility: The proposed method requires specialized attention masks, which may be incompatible with highly optimized attention kernels like FlashAttention. This raises concerns regarding the actual wall-clock speedups achieved in a production environment compared to standard sequence reduction methods. Instead, simple downsampling methods mentioned above do not have this issue and is compatible with FlashAttention training.

[1] Zhang, P., Liu, Z., Xiao, S., Shao, N., Ye, Q., & Dou, Z. (2024). Long Context Compression with Activation Beacon. International Conference on Learning Representations.
[2] Burchi, M., & Vielzeuf, V. (2021). Efficient Conformer: Progressive Downsampling and Grouped Attention for Automatic Speech Recognition. 2021 IEEE Automatic Speech Recognition and Understanding Workshop (ASRU), 8-15.

---

### Official Review · Reviewer_Jwsz · 2026-03-12

**Soundness:** 2
**Presentation:** 3
**Significance:** 2
**Originality:** 3
**Overall Recommendation:** 2
**Confidence:** 5

**Summary:**

This paper studies long-form speech understanding in speech-language models and proposes Speech-XL, a framework designed to improve the efficiency of processing long audio sequences. The central motivation is that speech encoders produce much longer token sequences than text, which leads to large attention cost and KV-cache memory growth when such sequences are fed into large language models. To address this, the paper introduces a Speech Summarization Token (SST) mechanism that compresses speech representations inside the LLM by summarizing local intervals of speech tokens into compact summary states. After an interval is processed, the model keeps the SST-based summary states and discards the dense original speech-token states, so later intervals can condition on compressed summaries instead of the full previous audio.

The architecture consists of a Whisper-small speech encoder, a dual-adapter bridge with separate semantic and acoustic branches, and a Qwen2.5-7B-Instruct LLM backbone augmented with SST compression. Training is staged in three phases: an ASR stage to establish speech–text alignment, a speech question-answering stage to incorporate richer acoustic and paralinguistic understanding, and a final compression stage where SST tokens are trained to summarize long-form audio. The paper evaluates Speech-XL on LongSpeech and AudioMarathon, and reports that the compressed model performs strongly on long-form speech tasks while achieving better inference efficiency on long audio.

**Compliance With Llm Reviewing Policy:**

Affirmed.

**Key Questions For Authors:**

See weaknesses.

**Limitations:**

Yes

**Strengths And Weaknesses:**

The authors study an important problem: scaling speech-language models to long-form audio while controlling memory and computation. The proposed SST mechanism is intuitive: instead of storing the full KV cache for all past speech tokens, the model learns compact summary tokens that retain important information while reducing the effective sequence length. I also think it is helpful that the paper includes ablation studies on compression strategies and curriculum learning, since those are among the more controlled parts of the empirical study.

However, I have substantial concerns about the experimental design and the strength of the empirical claims. The most serious issue is that the main benchmark comparisons are not controlled. For the LongSpeech results, the paper explicitly states: “It is necessary to clarify that Speech-XL undergoes fine-tuning on this specific dataset, while the compared models operate primarily in a zero-shot setting.” This substantially weakens Tables 1 and 2 as evidence of superiority over prior systems. In those tables, the comparisons do not isolate the proposed method; at best, they show that the authors’ overall system performs well after in-domain fine-tuning.

For the AudioMarathon comparison in Table 3, it is also problematic. The compared systems rely on very different foundation models, pretraining data scales, and architectures. The paper itself notes that the gap to stronger models may fundamentally reflect differences in foundational pretraining volume rather than the compression mechanism. That makes the table less useful for validating the proposed method itself.

Another weakness is that the architectural motivation is not fully developed. The authors strive to outline the concept of separating semantic and acoustic information via a dual-adapter bridge, but the paper does not clearly justify why the acoustic branch should specifically use a Q-former. The paper says that the acoustic adapter uses “2-layer Q-former blocks equipped with 64 trainable query vectors” to preserve acoustic-specific features such as tone and environment, but it does not explain why a Q-former is preferable to other possible connector designs, nor does it provide an ablation on that choice.

The efficiency comparison in Figure 4 is not ideally matched. Speech-XL is built on Qwen2.5-7B-Instruct, but Figure 4 compares it against Qwen2.5-Omni-7B. A more convincing comparison would be against a closer baseline, such as a vanilla speech-extended Qwen2.5-7B-Instruct system without SST, or a more closely matched Omni-based variant.

In terms of significance, the paper addresses an important problem, and the general direction is worthwhile. In terms of originality, the SST-based KV compression mechanism is an interesting design choice, but much of the broader system is built from familiar ingredients: Whisper, query-based adapters, staged training, and LLM-based speech understanding. My main issue is not that the idea lacks merit, but that the experimental evidence does not validate it in a sufficiently controlled way.

---

### Official Review · Reviewer_8MRG · 2026-03-12

**Soundness:** 2
**Presentation:** 3
**Significance:** 2
**Originality:** 2
**Overall Recommendation:** 2
**Confidence:** 4

**Summary:**

This paper presents Speech-XL, a framework designed to enable Large Speech Language Models (LSLMs) to handle long-form audio (e.g., sessions exceeding several minutes) efficiently. The core contribution is the Speech Summarization Token (SST) mechanism, which partitions the continuous speech feature sequence into fixed intervals and compresses the intra-interval information into a single SST's Key-Value (KV) states. By discarding the high-density original speech tokens and retaining only the SST KV cache, the model significantly reduces the memory footprint and computational complexity from quadratic to manageable levels. The authors also employ a three-stage curriculum learning strategy—progressing from ASR and Q&A to varied compression ratios (2x to 16x)—to maintain performance under high compression. Evaluations on LongSpeech and AudioMarathon benchmarks show that Speech-XL achieves competitive results with lower inference overhead compared to current baselines.

**Compliance With Llm Reviewing Policy:**

Affirmed.

**Key Questions For Authors:**

1. In Table 3, the 'Upper-bound' performance only provides a total score without breakdown metrics. This lack of granularity makes it difficult to assess which specific capabilities (e.g., content extraction vs. speaker modeling) are most affected by the SST compression compared to an uncompressed ceiling.
2. The paper lacks clarity regarding the final effective frame rate (sampling rate) of the speech tokens after SST compression. Is the resulting rate 6.25 Hz (assuming a 50Hz initial rate divided by 8)? This is a critical hyperparameter for reproducibility and for understanding the acoustic resolution being fed into the LLM.

**Limitations:**

The authors did not discuss limitations.

**Strengths And Weaknesses:**

## Strengths
1. Unlike traditional methods that rely on heavy downsampling or text-based mediation, this work explores a KV-space compression approach. Using a dedicated token (SST) to "summarize" interval information within the LLM’s latent space is a technically sound and elegant solution for the long-context bottleneck.
2. The use of Curriculum Learning to gradually increase the compression ratio is a well-motivated design choice. It effectively bridges the gap between high-fidelity short audio perception and high-compression long audio reasoning, providing a blueprint for training stable LSLMs.
3. The paper goes beyond standard ASR metrics, testing on challenging benchmarks like LongSpeech and AudioMarathon. The results demonstrate the model's ability to maintain paralinguistic sensitivity (e.g., emotion, speaker counting) even at high compression ratios, which is often a weak point for long-form models.

## Weaknesses
1. The effectiveness of the SST (Speech Summarization Token) mechanism is somewhat undermined by the overall performance gap between the proposed model and current state-of-the-art (SOTA) Large Speech Language Models (e.g., Qwen-Omni). Since the core claim is the "intrinsic KV sparsification capacity" of LLMs, the authors should demonstrate that this compression technique can be integrated as a post-training enhancement for top-tier LSLMs without significant performance degradation.
2. Although the paper claims computational efficiency, the inference pipeline introduces a significant architectural bottleneck. The SST mechanism requires waiting for an entire interval to be encoded before KV states can be discarded and the next chunk processed. This essentially turns the model into a chunk-level streaming system, which inherently introduces fixed latency. In contrast, standard models can utilize parallel pre-filling for the entire sequence. The authors should provide a detailed analysis of the Time-to-First-Token (TTFT) and compare the wall-clock time (not just TFLOPs) against models that use a single pre-fill pass.

---

### Decision · Program_Chairs · 2026-04-30

**Decision:**

Reject

**Comment:**

Overview of the paper: The paper introduces Speech-XL, a framework designed to handle long-form audio in Large Speech Language Models (LSLMs) by mitigating context length limitations and memory bottlenecks. The primary contribution is the Speech Summarization Token (SST) mechanism, which partitions speech sequences and compresses intra-interval acoustic information into a reduced set of Key-Value (KV) states. The authors implement a three-stage curriculum learning strategy to train this module, progressively advancing from low-ratio to high-ratio compression. The resulting model is evaluated on the LongSpeech and AudioMarathon benchmarks to measure its performance and inference efficiency.

Strengths:
- The approach of utilizing a dedicated token for KV space compression, combined with a progressive curriculum learning strategy, is a well-motivated method for addressing memory constraints in long-context audio processing.
- The proposed SST mechanism demonstrably reduces memory usage and TFLOPs during inference for extended audio sequences, offering practical computational resource savings.

Areas for improvement:
- Uncontrolled Baseline Comparisons: A critical issue raised by multiple reviewers is the lack of strictly controlled experimental baselines. Specifically, Speech-XL undergoes fine-tuning on the LongSpeech dataset while the compared baseline models are evaluated primarily in a zero-shot setting, making it difficult to disentangle the performance advantages of the SST mechanism from benchmark-specific adaptation.
- Limited Novelty and Missing Simple Baselines: The core SST approach closely parallels existing context compression techniques in the broader ML field (e.g., Activation Beacon) and lacks substantial architectural innovation tailored specifically to the unique properties of continuous speech. Also, the evaluation omits necessary comparisons against simpler, standard alternatives like progressive downsampling.
- Insufficient Architectural Justification and Mechanistic Analysis: The architectural design choices, such as utilizing a Q-former specifically for the acoustic branch, are not adequately justified or ablated. Additionally, the paper lacks detailed diagnostic experiments to analyze what specific types of information (e.g., lexical, prosodic, or speaker identity) the SSTs actually retain at higher compression ratios.
- Inference Latency and Hardware Compatibility Concerns: The inference pipeline introduces an architectural bottleneck because the SST mechanism requires an entire interval to be encoded before the next chunk can be processed, introducing fixed latency compared to models that utilize parallel pre-filling. Additionally, the requirement for custom attention masks raises practical concerns regarding compatibility with highly optimized attention kernels like FlashAttention.